# Water Footprint Assessment of Rainfed Crops with Critical Irrigation under Different Climate Change Scenarios in SAT Regions

Konda Sreenivas Reddy *, Vegapareddy Maruthi, Prabhat Kumar Pankaj ⓘ, Manoranjan Kumar ⓘ, Pushpanjali, Mathyam Prabhakar, Artha Gopal Krishna Reddy, Kotha Sammi Reddy, Vinod Kumar Singh and Ashishkumar Kanjibhai Koradia

ICAR-Central Research Institute for Dryland Agriculture (CRIDA), Santhoshnagar, Hyderabad 500059, India; v.maruthi@icar.gov.in (V.M.); pankaj.pk@icar.gov.in (P.K.P.); manoranjan.kumar@icar.gov.in (M.K.); pushpanjali@icar.gov.in (P.); m.prabhakar@icar.gov.in (M.P.); agk.reddy@icar.gov.in (A.G.K.R.); k.sammireddy@icar.gov.in (K.S.R.); director.crida@icar.gov.in (V.K.S.); akkoradia51@gmail.com (A.K.K.)
* Correspondence: ks.reddy@icar.gov.in

**Abstract:** Semi-Arid Tropical (SAT) regions are influenced by climate change impacts affecting the rainfed crops in their productivity and production. Water Footprint (WF) assessment for rainfed crops on watershed scale is critical for water resource planning, development, efficient crop planning, and, better water use efficiency. A semi-arid tropical watershed was selected in lower Krishna river basin having a 4700 ha area in Telangana, India. Soil and Water Assessment Tool (SWAT) was used to estimate the water balance components of watershed like runoff, potential evapotranspiration, percolation, and effective rainfall for base period (1994 to 2013) and different climate change scenarios of Representative Concentration Pathways (RCP) 2.6, 4.5 and 8.5 for the time periods of 2020, 2050 and 2080. Green and blue WF of rainfed crops viz., maize, sorghum, groundnut, redgram and cotton were performed by considering rainfed, and two critical irrigations (CI) of 30 mm and 50 mm. It indicated that the effective rainfall (ER) is less than crop evapo-transpiration (ET) during crop growing period under different RCPs, time periods, and base period. The green WF under rainfed condition over different RCPs and time periods had decreasing trend for all crops. The study suggested that in the rainfed agro-ecosystems, the blue WF can significantly reduce the total WF by enhancing the productivity through critical irrigation management using on farm water resources developed through rainwater harvesting structures. The maximum significant reduction in WF over the base period was observed 13–16% under rainfed, 30–32% with 30 mm CI and 40–42% with 50 mm CI by 2080. Development of crop varieties particularly in oilseeds and pulses which have less WF and higher yields for unit of water consumed could be a solution for improving overall WF in the watersheds of SAT regions.

**Keywords:** green and blue water footprint; crop evapotranspiration; effective rainfall; rainfed crops; climate change; watershed

## 1. Introduction

Natural resources, particularly water and food supply, are at tremendous pressure due to global population rise and dynamic changes in the consumption pattern of society, and India, which is projected to be the world's most populated country by 2027, will be one of the most impacted countries [1]. This will have a direct impact on water and land resource availability vis-à-vis agriculture. It is predicted that severe water scarcity is affecting one billion population in India at least for one month of the year which stresses the need for efficient water resource development and management [2]. Rainfed (green water) farming systems in Semi-Arid Tropical (SAT) regions provide diverse food supplies from 51% of net sown area (139.4 mha) in India [3]. SAT regions contribute 60% of nutritive food grains,

although is suffering with 20 to 35% undernourished population [4]. As per IPCC report (AR5), the climate change impacts would lead to global warming by increased temperature from 2 to 5 °C by the end of the century with increased extreme weather events [5]. Indian agriculture is also affected by changes in the rainfall pattern, high intense rainfall, floods, and droughts contributing to the overall reduction in the crop productivity, soil quality, and accelerated land degradation due to erosion, availability of both blue and green water, etc., in the SAT regions. The increase in extreme weather events can affect the crop productivity in the SAT regions of India which contributes to the production of cereals, pulses, oil seeds, cotton, etc., under rainfed farming [6]. Extreme weather events are the greatest global risk in the present climate change [1]. The global requirement of cereals would increase by 55–80% by the year 2050 which can be accomplished through expansion of area under crop or by increasing crop productivity since land and water resources are limited [7].

Agriculture is the highest consumer of global fresh water at 70%. However, India accounts for 80% of fresh water consumption in agriculture [8]. Rainwater harvesting is one of the best options considered in the SAT regions of India for improving the water productivity in the diversified cropping system with improved benefits to the farmer [6]. The Lower Krishna river basin of Telangana, India is of 25.8 million ha, which contributes to a major irrigation project of Nagarjuna Sagar dam. Integrated watershed management programmes are implemented extensively in the region and have the scope for improvement in the water resource development and efficient utilization to manage dry spells [9]. The crop water balance analysis for maize and cotton in the SAT regions indicated that there was decrease in the seasonal rainfall in the normal sowing window and increase in crop water requirements by 2050 for maize and cotton [10]. Water storage on farm provides a mechanism for dealing with the variability in rainfall which, if planned and managed efficiently, increases water security, agricultural productivity, and adaptive capacity to climate change [11].

Water footprint (WF) within the agricultural sector has been extensively studied, mainly focusing on the water footprint of crop production. The WF of domestic, industrial, and agricultural sectors has been calculated and reported at the sub-national region level [12,13] as well as at the national level [14–19] and the global level [20–23]. The green and blue water footprint of crop production are estimated by using a grid-based dynamic water balance model considering local climate and soil conditions after calculating the effective rainfall (ER), potential evapotranspiration (PET), and crop water requirements. Most of these studies pertain irrigated eco systems under major irrigation systems which are different from SAT regions that are critically rainfall dependent. Due to weather aberrations in the SAT region with long dry spells during crop growth stages, there is a need to critically analyse the water supplies for rainwater harvesting on farms and its utilization during dryspells at critical stages of crop growth and its impacts on water footprints for rainfed crops in watersheds [24]. Therefore, a Water footprint assessment would help to make a policy framework for the adaptation of climate-resilient technologies, particularly rainwater harvesting through on-farm reservoirs and efficient use of water resources in the rainfed region on a watershed basis [6,25–27].

The Soil and Water Assessment Tool (SWAT) was used for estimating the runoff, potential evapotranspiration, and percolation apart from other components of groundwater recharge. The rainfall effectiveness (green water use) was evaluated for different crops in the Nagarjuna Sagar canal command area of Andhra Pradesh using SWAT [28]. The spatial optimization of soil and water conservation practices was studied on a watershed scale using SWAT and evolutionary algorithm [27]. The blue and green proportions of crop ET of six important crops were quantified [29] and four major land-use types of Kothakunta sub watershed in Andhra Pradesh for water footprint assessment on a basin scale. The water footprint for 15 different crops was estimated at basin level in the Indo-Gangetic region [30]. The green, blue and grey Water footprint of 126 crops all over the world for the period 1996–2005 was estimated with a high spatial resolution [22]. Various studies on water footprint for different climate change scenarios were reported using different downscaling

models which are region-specific, particularly for irrigated rice [31,32]. Many studies have been reported representing the impacts of climate change the at global and regional levels for irrigated crops on a basin scale. The present study focused on WF assessment for rainfed crops on the watershed scale in SAT regions with adaptation strategies of rainwater harvesting through on-farm reservoirs in a watershed.

## 2. Material and Methods

### 2.1. Study Area and Climate

The present study was conducted in a watershed consisting of 8 tribal villages of Padara Mandal, Nagarkurnool district of Telangana state (Figure 1). The area lies between 16°27′ N and 79°1′ E. The watershed has its automatic weather station in PadaraMandal. The watershed having an area of 4700 ha was delineated into several sub watersheds with different land use, soil characteristics and slopes. According to the 20 years observation data, the average annual rainfall in the watershed is 734 mm, of which the average south west seasonal rainfall accounts for 86%. Two-thirds of the rainfall occurs during the period of July to October. The average maximum and minimum temperatures of the area are 33 °C and 12 °C. The elevation of the selected area is 145 m above mean sea level.

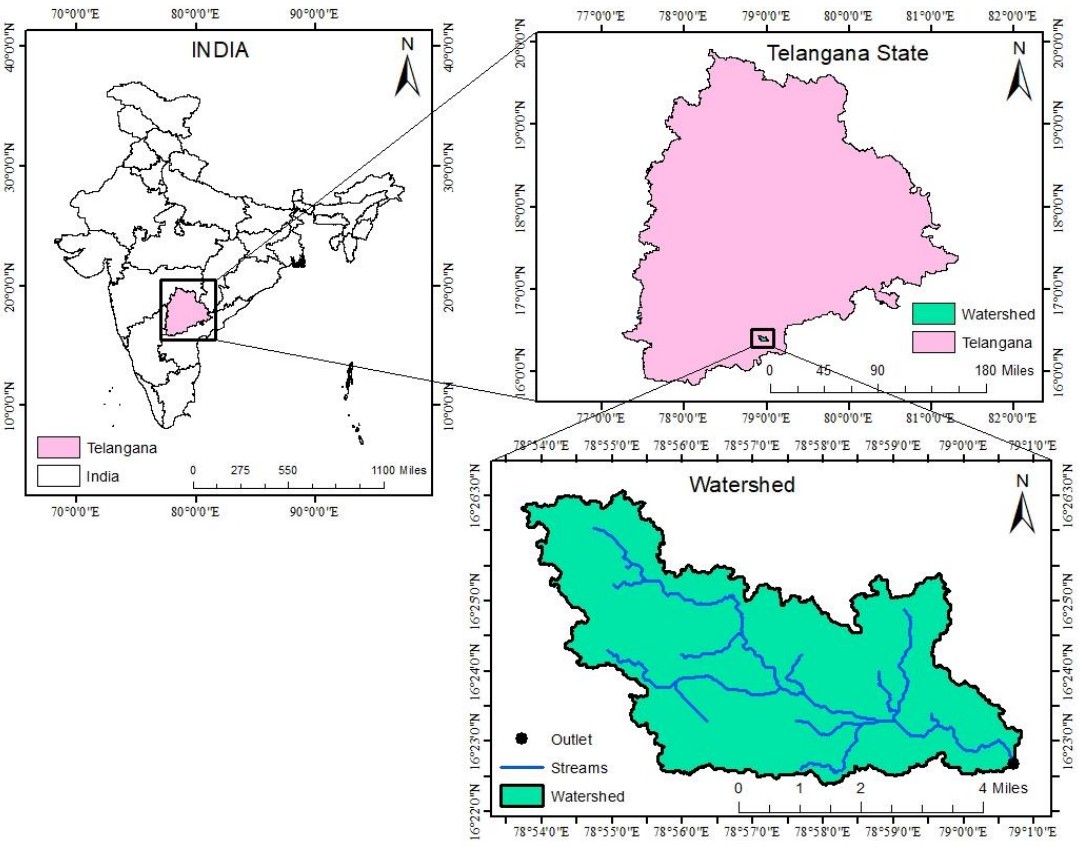

**Figure 1.** Location map of the watershed.

The area is dominated by sandy clayey loam soils accounting for around 78% with poor soil health. The major land use consists of agriculture (31%) and small bushes (56%) and forest (7.6%). Agriculture in the watershed mainly consists of seasonal rainfed crops like maize, cotton, redgram, groundnut, and sorghum. The watershed has a rolling topography having slopes from 1–11% on average.

### 2.2. Data Acquisition

Data required for the study were compiled from different sources. Digital elevation map from the ASTER Satellite with an accuracy of 30 m was obtained from USGS. Land

use/land cover map was obtained from IRS-LISS III. Spatial distribution of major soil types and sand, silt, and clay content of these soils were taken from grid-based Harmonized World Soil Database (HWSD-FAO) [33]. Crop coefficients at different crop development stages (initial, middle, and late-stage), were taken from FAO report [34].

Climate Data

The climate data related to monthly average rainfall and temperature of the study area under various RCPs and time periods are presented in Figure 2.

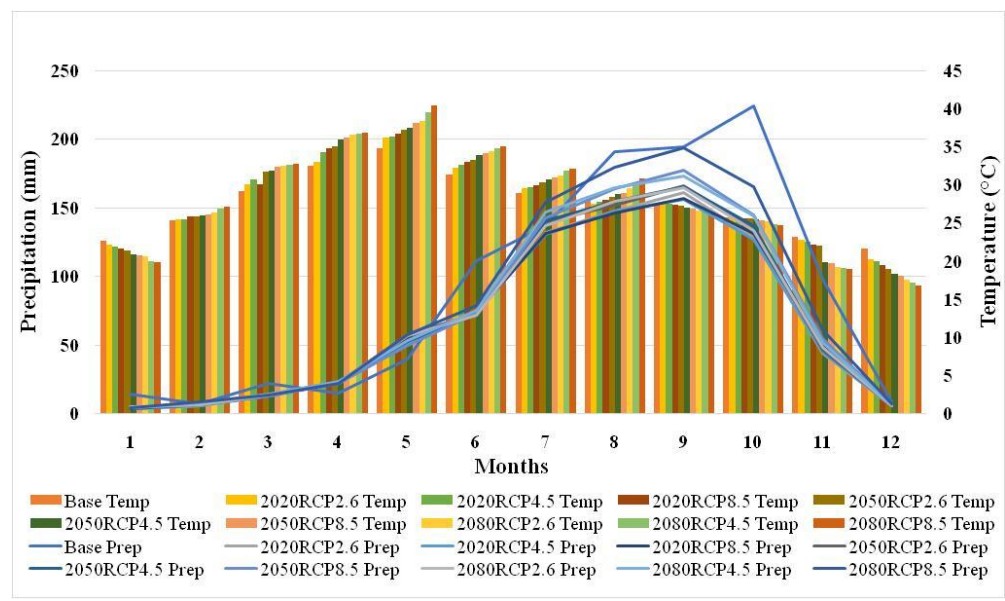

**Figure 2.** Monthly distribution of average precipitation (Prep) and temperature (Temp) for different RCP, base period, and time periods.

### 2.3. Estimation of Water Balance in a Watershed Using SWAT

Soil Water Assessment Tool (SWAT) was used to estimate the watershed surface runoff, potential evapotranspiration, and percolation rates, stream flows, etc. In this model, a watershed is divided into multiple sub-watersheds that are then further subdivided into unique soil/land use characteristics called hydrologic response units (HRUs) using ARC-GIS. The input layers of daily rainfall, temperature, relative humidity, radiation, wind, DEM, Land use, Soil cover, etc. were provided to the model (Figure S1). Digital elevation model (DEM) at a resolution of $30 \times 30$ m was used as input for delineation. In this study, three emission scenarios from the IPCC were used as RCP 2.6, 4.5, and 8.5, representing low, medium, and high radiant energy levels. The long-term data on runoff was generated for three climate change scenarios of 2020, 2050, and 2080. Using these data, effective rainfall of different rainfed crops, PET, and crop ET ($ET_c$) were calculated.

### 2.4. Estimation of Crop Yields in Different Climate Change Scenarios

The crop yields for selected crops namely maize, sorghum, groundnut, redgram, and cotton were assessed using the AquaCrop using soil, climate, crop, and water use data under climate change scenarios of RCP 2.6, 4.5, and 8.5 for different time periods of 2020, 2050, and 2080 including base period. The crop yields were estimated for rainfed and two critical irrigation (CI) levels of 30 mm and 50 mm at critical stages of crops for climate change scenarios and base period. Based on the experience, the two levels of 30 mm and 50 mm for both deficit and intensive critical irrigations were found optimum for SAT regions with sandy clay loam soils [6]. These data were used for calculating the WF of selected crops.

*2.5. Water Footprint (WF) Assessment*

The base data on crops, and existing land productivity in a watershed were taken to calculate both green and blue water footprints. The blue and green *WF* (*WF_{blue}* and *WF_{green}*) of rainfed crops were calculated based on the standard methods proposed in the Water Footprint Assessment Manual [23]. Presently, all the crops are grown under rainfed which is primarily rainfall-dependent production. As the scope for runoff water harvesting is seen in the selected watershed, two critical irrigations of 30 mm and 50 mm were taken to provide as a drought management strategy for improving the yields in the existing rainfed cropping systems of the watershed. Using this information, the water footprints were calculated for the existing crops in a watershed. However, the grey water footprint is neglected in the watershed due to the very low application rate of fertilizers in the rainfed agriculture in the watershed by the farmers.

2.5.1. Green Water Footprint

The green crop water use (*CWU_{green}*) is estimated by considering two parameters namely crop evapotranspiration (*ET_c*) and Effective rainfall (*P_{eff}*) during crop growth period. Minimum of these values is considered for calculating the water footprint as given below:

$$CWU_{green} = 10 \times \sum Min\left(P_{eff}, ET_c\right) \tag{1}$$

*ET_c* was calculated by using crop coefficients at different growth stages of selected crops in the watershed as given below

$$ET_c = ET_0 \times K_c \tag{2}$$

where *ET_0* is potential evapotranspiration (mm) and *K_c* is crop coefficient. Crop coefficients (*K_c*) were obtained from FAO [35]. Crop planting dates and lengths of cropping seasons were obtained from PJTSAU [36] (Table 1).

**Table 1.** Crop characteristics of different crops.

| Crop | $K_{c\_Ini}$ | $K_{c\_mid}$ | $K_{c\_end}$ | Date of Sowing | Length of Crop Growing Period (Days) |
|---|---|---|---|---|---|
| Maize | 0.3 | 1.2 | 0.5 | 05-July | 120 |
| Sorghum | 0.3 | 1 | 0.55 | 05-July | 115 |
| Groundnut | 0.4 | 1.15 | 0.6 | 10-July | 120 |
| Cotton | 0.35 | 1.2 | 0.6 | 15-July | 180 |
| Redgram | 0.3 | 1 | 0.5 | 10-July | 120 |

Source: $K_c$ values: FAO (1988) [35], Sowing dates and length of crop periods: PJTSAU (2019) [36].

Effective rainfall for different crops was calculated by using USDA [37] as given below:

$$P_{eff} = R - SR_0 - PR \tag{3}$$

where *R* is daily rainfall (mm), *SR_0* is surface runoff (mm) and *PR* is percolation (mm).

2.5.2. Blue Water Footprint

The blue crop water use (*CWU_{blue}*) is the amount of surface and groundwater used by the crop over the entire crop growing period i.e., the amount of water provided as critical irrigation (*I_c*) in addition to effective rainfall to the crop during the growing period. The total green crop water use is the summation of *ET_c* or *P_{eff}* over the crop growth period.

$$CWU_{blue} = 10 \times \sum I_c \tag{4}$$

The crop water uses over the crop growing period (m$^3$ ha$^{-1}$) were obtained by multiplying with factor 10 which converts water depths (mm) into water volumes per unit surface area (m$^3$ ha$^{-1}$). The green water footprint ($WF_{green}$, m$^3$/t) and blue water footprint ($WF_{blue}$, m$^3$/t) were calculated by dividing the green crop water use ($CWU_{green}$) and the blue crop water use ($CWU_{blue}$) by the yield of different crops respectively [23] as given below:

$$WF_{green} = CWU_{green}/Y_r \tag{5}$$

$$WF_{blue} = CWU_{blue}/Y_{I_c} \tag{6}$$

The total water footprint of a crop ($WF$, m$^3$/t) is the sum of the green and blue components:

$$WF = WF_{green} + WF_{blue} \tag{7}$$

### 2.5.3. Water Footprints for Climate Change Scenarios

The long-term data on *SRO*, *PET* and *PR* for different RCPs were simulated using SWAT for the periods of 2020, 2050, and 2080. Crop evapotranspiration and effective rainfall were calculated for the scenarios of 2.6, 4.5, and 8.5 for future time periods of 2020, 2050, and 2080. The *WF* for climate change scenarios of RCP 2.6, 4.5 and 8.5 for the periods of 2020, 2050 and 2080 were calculated by providing downscaled rainfall obtained from global climate models (GCM) as input to the calibrated SWAT model. It was assumed that there will be no change in land use for the project area in the future. Climate variables in the future, such as wind speed, relative humidity, and sunshine hours, were also assumed to be the same as that of the base period.

### 3. Results

*3.1. SWAT Calibration and Validation*

Calibration and validation were carried out using SWAT-CUP. Sensitive parameters were identified for the selected watershed in the first step. Then the model parameters were calibrated on daily basis comparing the observed and simulated runoff values in the watershed. The validated results are presented in Figure 3 with an R$^2$ (Coefficient of determination) of 0.87. It indicated that there is a close relationship between observed and simulated runoff in a watershed and the model can be applied to the watershed considered under the present study.

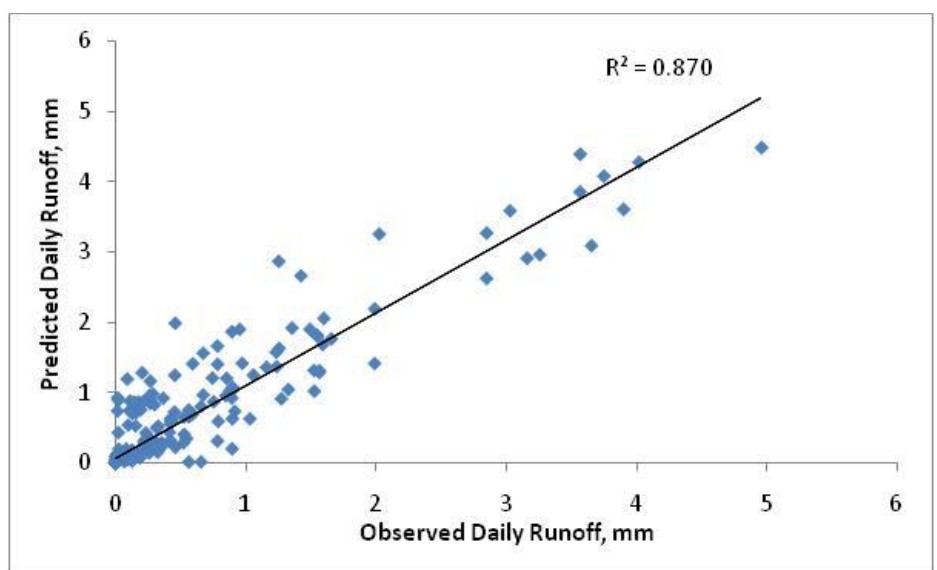

**Figure 3.** Validation of SWAT model in watershed.

### 3.2. Water Balance

SWAT outputs of surface runoff, potential evapotranspiration and percolation were taken for calculating effective rainfall (ER), crop evapotranspiration ($Et_c$) for all selected rainfed crops during their crop growth period. These calculations were made for RCPs (2.6, 4.5 and 8.5) with different time periods (2020, 2050, and 2080). The total rainfall during crop growth stages of different crops has an increasing trend in RCPs and time periods over the base period (Figure 4). The percentage increase in the total rainfall varied from 1.48–2.26% in RCP2.6 in 2020 and 23.5–26.85% in RCP 8.5 in 2080 across the crops over the base period. Accordingly, the surface runoff also increased from 68 mm to 121 mm in maize, 73 mm to 122 mm in sorghum, 74 mm to 124 mm in groundnut, 120 mm to 240 mm in redgram and 119 mm to 200 mm in cotton across the RCPs and time periods. The maximum surface runoff was found under the RCP 8.5 by 2080. During the base period, the surface runoff was found less varying from 68 mm to 114 mm across the crops. The analysis indicated that there was potential for rainwater harvesting through on-farm reservoirs for implementing critical irrigation in watershed for selected crops in both base period and RCPs and time periods of 2020 to 2080. The effective rainfall was found less than $ET_c$ in base period as well as in RCPs for different time periods. The ER was taken for calculating green WF for the respective crops. The percolation varied from 52 mm to 96 mm in different crops and the maximum was noticed in deep-rooted crops like redgram and cotton.

### 3.3. Water Footprint of Rainfed Crops

3.3.1. Base Period (1994–2013)

The analysis was carried out for five rainfed crops commonly grown in the selected watershed for a base period of 30 years. The crops considered are maize, sorghum, groundnut, redgram and cotton. The crop ET and ER were calculated from SWAT water balance. All the rainfed crops are considered with two critical irrigations during kharif at two critical stages of crops. The average estimated crop ET and effective rainfall for their growing period are presented in Table 2 for base period and for different climate change scenarios. It is observed that the ER was less than crop ET for all the rainfed crops. Green water footprint was calculated by taking a minimum of crop ET and effective rainfall for all the rainfed crops. The average simulated crop yields through AquaCrop are presented in Table 3 for different climate change scenarios and time periods and for the base period.

**Table 2.** Crop ET and effective rainfall of different crops during crop growth period.

| | Maize | | Sorghum | | Groundnut | | Redgram | | Cotton | |
|---|---|---|---|---|---|---|---|---|---|---|
| | $ET_c$ (mm) | ER (mm) | $ET_c$ (mm) | ER (mm) | $ET_c$ (mm) | ER (mm) | $ET_c$ (mm) | ER (mm) | $ET_c$ (mm) | ER (mm) |
| Base period | 464.57 | 321.93 | 455.20 | 331.88 | 508.09 | 319.74 | 455.33 | 352.98 | 705.55 | 401.62 |
| 2020-2.6 | 465.63 | 322.62 | 462.63 | 332.14 | 510.16 | 319.92 | 456.82 | 353.24 | 707.13 | 401.70 |
| 2020-4.5 | 466.72 | 323.90 | 463.74 | 333.92 | 511.09 | 321.74 | 457.62 | 354.84 | 708.59 | 402.15 |
| 2020-8.5 | 467.68 | 325.73 | 464.25 | 334.95 | 512.88 | 322.61 | 458.12 | 356.65 | 709.14 | 403.12 |
| 2050-2.6 | 469.87 | 329.26 | 467.25 | 339.32 | 516.76 | 326.83 | 461.58 | 360.64 | 713.67 | 407.80 |
| 2050-4.5 | 473.55 | 330.54 | 468.76 | 340.86 | 517.61 | 327.57 | 462.18 | 361.62 | 714.37 | 408.96 |
| 2050-8.5 | 474.16 | 332.66 | 470.47 | 342.64 | 521.77 | 330.44 | 464.32 | 364.33 | 718.95 | 413.33 |
| 2080-2.6 | 471.59 | 331.99 | 469.69 | 341.02 | 518.72 | 328.59 | 462.07 | 362.08 | 715.73 | 409.46 |
| 2080-4.5 | 475.70 | 334.16 | 470.50 | 345.60 | 521.09 | 331.43 | 463.48 | 365.68 | 718.01 | 413.83 |
| 2080-8.5 | 478.37 | 339.06 | 476.18 | 350.48 | 532.72 | 338.06 | 470.31 | 370.97 | 728.29 | 421.82 |

$ET_c$ = Crop Evapotranspiration, ER = Effective Rainfall.

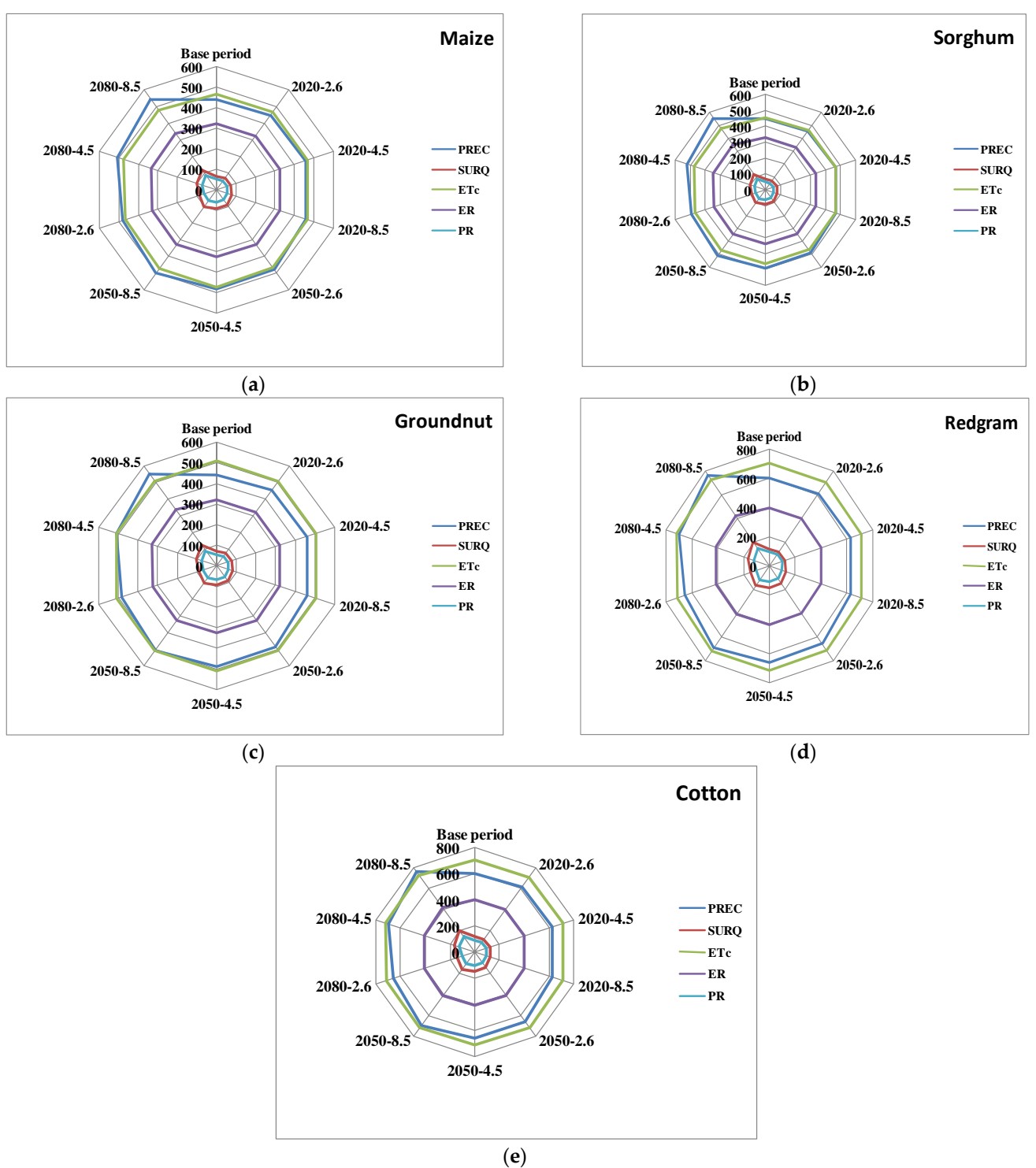

**Figure 4.** SWAT estimated water balance (mm) for selected crops during their growing period for the base period and climate change scenarios. (PREC, Precipitation; SURQ, Surface runoff; ET$_c$, Crop evapotranspiration; ER, Effective rainfall; PR, Percolation). (**a**) Maize, (**b**) Sorghum, (**c**) Groundnut, (**d**) Redgram, (**e**) Cotton.

**Table 3.** Simulated crop yields under rainfed and critical irrigations using AquaCrop.

| | Maize (t/ha) | | | Sorghum (t/ha) | | | Groundnut (t/ha) | | | Redgram (t/ha) | | | Cotton (t/ha) | | |
|---|---|---|---|---|---|---|---|---|---|---|---|---|---|---|---|
| | Rainfed | 30 mm | 50 mm | Rainfed | 30 mm | 50 mm | Rainfed | 30 mm | 50 mm | Rainfed | 30 mm | 50 mm | Rainfed | 30 mm | 50 mm |
| Base period | 1.50 | 2.20 | 4.20 | 1.00 | 1.76 | 2.98 | 1.30 | 2.70 | 4.80 | 0.80 | 1.51 | 2.60 | 0.90 | 2.40 | 4.50 |
| 2020-2.6 | 1.53 | 2.29 | 4.51 | 1.02 | 1.84 | 3.17 | 1.34 | 2.90 | 5.19 | 0.82 | 1.62 | 2.80 | 0.93 | 2.58 | 4.87 |
| 2020-4.5 | 1.56 | 2.39 | 4.74 | 1.05 | 1.91 | 3.35 | 1.36 | 2.99 | 5.49 | 0.85 | 1.66 | 2.94 | 0.98 | 2.66 | 5.17 |
| 2020-8.5 | 1.60 | 2.51 | 4.98 | 1.09 | 2.05 | 3.61 | 1.39 | 3.17 | 5.85 | 0.87 | 1.76 | 3.14 | 0.99 | 2.80 | 5.51 |
| 2050-2.6 | 1.65 | 2.62 | 5.25 | 1.12 | 2.11 | 3.92 | 1.44 | 3.29 | 6.27 | 0.90 | 1.82 | 3.39 | 1.01 | 2.92 | 5.91 |
| 2050-4.5 | 1.70 | 2.76 | 5.56 | 1.16 | 2.21 | 4.19 | 1.49 | 3.41 | 6.63 | 0.92 | 1.89 | 3.65 | 1.04 | 3.04 | 6.33 |
| 2050-8.5 | 1.74 | 2.86 | 6.05 | 1.18 | 2.31 | 4.42 | 1.52 | 3.59 | 7.05 | 0.94 | 1.98 | 3.88 | 1.06 | 3.21 | 6.78 |
| 2080-2.6 | 1.76 | 2.94 | 6.42 | 1.20 | 2.35 | 4.73 | 1.55 | 3.68 | 7.58 | 0.95 | 2.04 | 4.09 | 1.08 | 3.27 | 7.09 |
| 2080-4.5 | 1.80 | 3.05 | 6.97 | 1.21 | 2.46 | 5.00 | 1.58 | 3.84 | 7.98 | 0.98 | 2.11 | 4.32 | 1.10 | 3.39 | 7.50 |
| 2080-8.5 | 1.83 | 3.15 | 7.30 | 1.24 | 2.51 | 5.20 | 1.61 | 3.92 | 8.47 | 1.00 | 2.18 | 4.58 | 1.13 | 3.52 | 8.00 |

The yields of maize were 1.5 t/ha, 2.2 t/ha and 4.2 t/ha for rainfed, 30 mm and 50 mm critical irrigations in the base period, respectively. The effective rainfall for maize was 321.93 mm as compared to crop ET of 464.6 mm. The green WF for maize was 2146 $m^3$/t and blue WFs were 273 $m^3$/t and 238 $m^3$/t for 30 mm and 50 mm CI, respectively (Figure 5). The yields for the sorghum were 1 t/ha, 1.76 t/ha and 2.98 t/ha under rainfed, with two critical irrigations of 30 mm and 50 mm, respectively. Effective rainfall for the base period was 331.88 mm with a green WF of 3319 $m^3$/t for sorghum. The blue WFs for sorghum were 341 $m^3$/t and 336 $m^3$/t under 30 mm and 50 mm CI, respectively.

The ER calculated during the growing period for groundnut crop was 319.7 mm against crop ET of 508 mm. The yields of groundnut were 1.3 t/ha, 2.7 t/ha and 4.8 t/ha for rainfed, 30 mm, and 50 mm CI, respectively. The green WF for groundnut was 2460 $m^3$/t and the blue WFs were 222 $m^3$/t and 208 $m^3$/t under 30 mm and 50 mm CI's, respectively (Figure 5). The ER during the growing period of redgram was 353 mm against crop ET of 455 mm. The yields of redgram were 0.8 t/ha, 1.5 t/ha and 2.6 t/ha under rainfed, 30 mm, and 50 mm CI's, respectively. The green WF for redgram was 4412.25 $m^3$/t and blue WF's were 397 $m^3$/t and 384 $m^3$/t for 30 mm and 50 mm CI's, respectively. The effective rainfall for cotton during its growth period was 401.6 mm as compared to crop ET of 705.6 mm.

The yields of cotton were 0.9 t/ha, 2.4 t/ha and 4.5 t/ha under rainfed, 30 mm, and 50 mm CI, respectively. The green WF was 4462.5 $m^3$/t and the blue WFs were 250 $m^3$/t at 30 mm and 222 $m^3$/t at 50 mm CI (Figure 5). The strategy of critical irrigations two times during crop season reduced the WF as compared to rainfed which totally depends on the utilization of ER as green water storage in the root zone. ER contribution to the crop yields is rainfall-dependent during the crop growing period. The total WF was minimum for maize as compared to all other crops under rainfed system indicating that the crop has better utilization of water converting into higher yields than the other crops followed by sorghum, groundnut, redgram and cotton.

3.3.2. Green and Blue Water Footprints of Rainfed Crops under Different Climate Change (CC) Scenarios

The green and blue water footprints were calculated for three CC scenarios of RCP 2.6, 4.5 and 8.5 for the time periods of 2020, 2050 and 2080 and the results are presented in Figure 5.

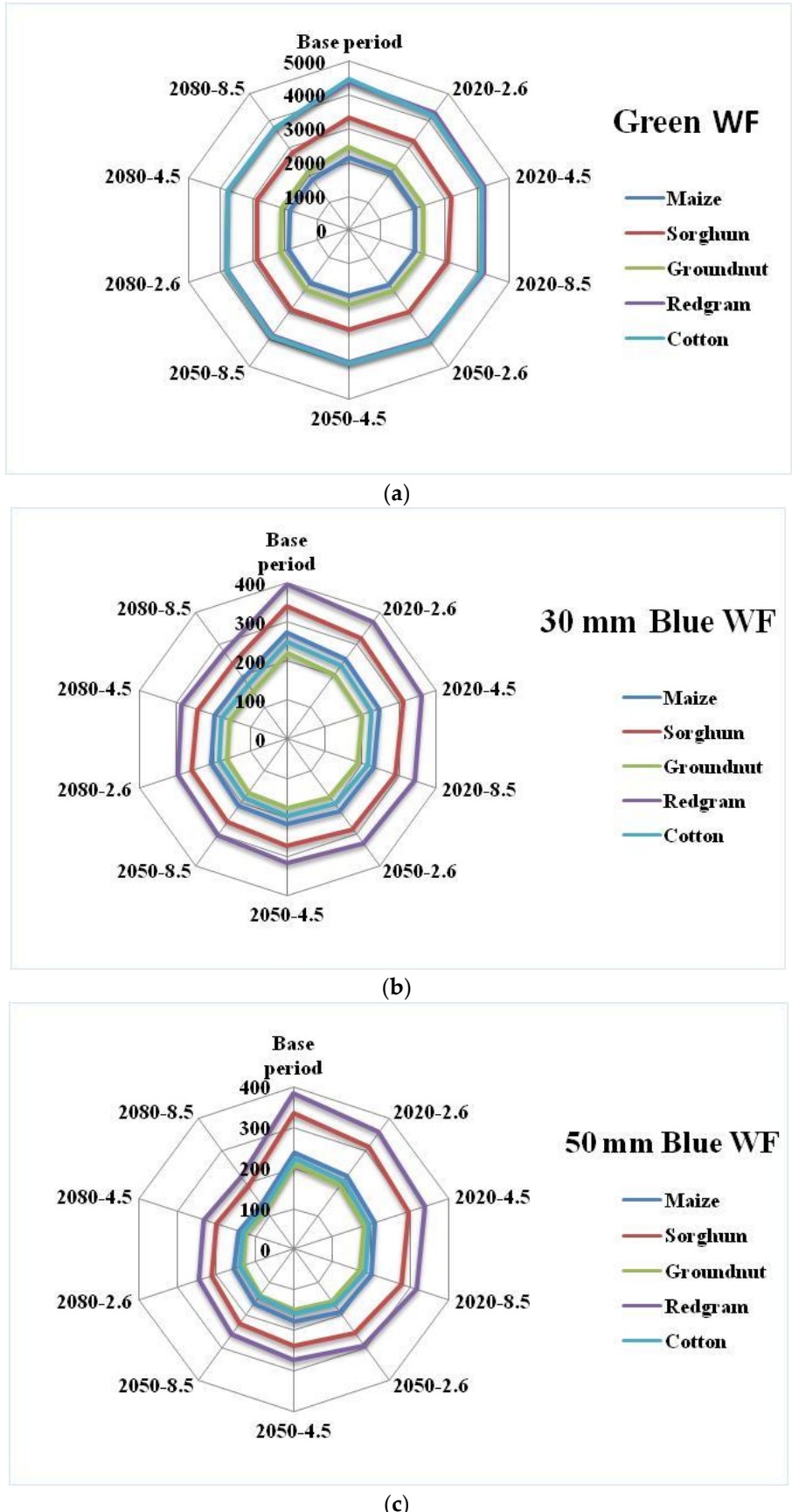

**Figure 5.** Water Footprint (m³/t) of rainfed crops with critical irrigation and different RCPs with time periods (**a**) Green WF (**b**) Blue WF CI: 30 mm (**c**) Blue WF CI: 50 mm.

Maize

The crop ET varied from 465.6 mm to 478.4 mm with an increasing trend during its crop growth period of 117 days in different scenarios with a minimum in RCP 2.6 in 2020 and a maximum in RCP 8.5 in 2080. The ER of maize varied from 322.6 mm to 339 mm with an increasing trend in different RCP scenarios. However, the ER observed was less in RCP scenarios for the time periods of 2020 to 2080. Green and blue WF were calculated by taking the yields of 1.5 t/ha, 2.2 t/ha and 4.2 t/ha in rainfed, 30 mm and 50 mm CI's strategies for the base period. There was a slight increase in the maize yield from 1.53 t/ha to 1.76 t/ha in RCP 2.6 for different time periods. Similar trend was observed in RCP 4.5 and 8.5 with a range from 1.56 t/ha to 1.8 t/ha and 1.6 t/ha to 1.83 t/ha under time periods. The green WF of maize reduced from 2106 $m^3$/t to 1886 $m^3$/t in RCP 2.6 during the time period of 2020 to 2080 (Figure 5a). In RCP 4.5, it reduced from 2074 $m^3$/t to 1862 $m^3$/t from2020 to 2080. In RCP 8.5, green WF varied from 2035 $m^3$/t to 1853 $m^3$/t. It was observed that there was a decrease of 1.88%, 7% and 12.1% of green WF in RCP 2.6 for different periods of 2020, 2050 and 2080 respectively over the base period (Figure 6a). In RCP 4.5 green WF was decreased by 3.4%, 9.3%, and 13.3% in different time periods (2020, 2050, and 2080) over the base period. In RCP 8.5, the maximum decrease in green WF (rainfed) was observed varying from 5.2–13.7% in different time periods as compared to the base period.

The WF of maize with two CI of 30 mm and 50 mm as an adaptation strategy to CC, the blue WF with 30 mm CI varied from 255 $m^3$/t to 204 $m^3$/t in RCP 2.6, 248 $m^3$/t to 195 $m^3$/t in RCP 4.5 and 235 $m^3$/t to 190.5 $m^3$/t in RCP 8.5 during the time period of 2020 to 2080 (Figure 5b). Similarly, with a 50 mm CI strategy, the blue WF varied from 222 $m^3$/t to 156 $m^3$/t in RCP 2.6, 209 $m^3$/t to 143 $m^3$/t in RCP 4.5 and 201 $m^3$/t to 137 $m^3$/t in RCP 8.5 during 2020 to 2080 (Figure 5c). Though the blue WF of maize has decreasing trend within RCPs from 2020 to 2080, the WF was decreased over rainfed (green WF). The percentage decrease in blue WF was 6.5–25%, 9–28.4% and 13.7–30.2% in RCP 2.6, 4.5 and 8.5 respectively for different time periods. In 50 mm CI strategy, the blue WF was further reduced by 6.9% to 35%, 12.3–39.7%, and 15.7–42.5% in RCP 2.6, 4.5, and 8.5, respectively (Figure 6a) for different time periods indicating the optimum adaptation strategy for maize in SAT regions.

Sorghum

The crop ET varied from 455 mm to 476.2 mm in different scenarios of climate change (RCP 2.6 to RCP 8.5) during the time periods of 2020 to 2080. Similarly, the ER for the sorghum varied from 332.1 mm to 350.5 mm which is less than crop ET. Therefore, ER is considered for calculating green WF for sorghum in different RCPs and time periods. The yields estimated in different RCPs and time periods varied from 1.02 t/ha to 1.24 t/ha under rainfed, 1.87 t/ha to 2.51 t/ha with 30 mm CI and 3.2 t/ha to 5.2 t/ha with 50 mm CI. The predicted yields had an increasing trend over the RCPs and time periods over the base period (Table 3). The green WF varied from 3253 $m^3$/t to 2849 $m^3$/t, 3168 $m^3$/t to 2849 $m^3$/t and 3067 $m^3$/t to 2838 $m^3$/t in RCP 2.6, 4.5 and 8.5 respectively for different time periods (Figure 5).The green WF decreased from 1.98% to 14.2%, 4.5% to 14.2% and 7.6% to 14.5% over the base period in RCP 2.6, 4.5 and 8.5 respectively over time periods of 2020, 2050 and 2080 (Figure 6b).

With the adaptation strategy of CI's with 30 mm, the blue WF varied from 321 $m^3$/t to 257.6 $m^3$/t, 314 $m^3$/t to 243 $m^3$/t, 293 $m^3$/t to 239 $m^3$/t in RCP 2.6, 4.5 and 8.5, respectively for different time periods (Figure 5b). The percentage decrease in blue WF for 30 mm CI was 5.88–24.4%, 8–28.7%, and 14.1–29.9% over the base period among different RCP and time periods (Figure 6b). In the case of 50 mm CI two times, the blue WF varied from 313.5 $m^3$/t to211.4 $m^3$/t, 296 $m^3$/t to 199.6 $m^3$/t and 277 $m^3$/t to 192 $m^3$/t in RCP 2.6, 4.5 and 8.5, respectively during the time period of 2020 to 2080 (Figure 5c). The maximum decrease was observed with a 50 mm CI strategy varying from 6.58–37%, 11.8–40.5% and 17.5–42.7% (Figure 6b) over the base period in different RCP and time periods. The analysis indicated that the rainfed sorghum when cultivated with effective rainfall, the WF's were

maximum as compared to the adaptation strategy of giving critical irrigations with 30 mm and 50 mm two times during its crop growth period. Among the blue WFs, 50 mm CI reduced maximum WF in all RCP scenarios and time periods. However, it was found that there was a decreasing trend with RCP and time periods in all crops WF.

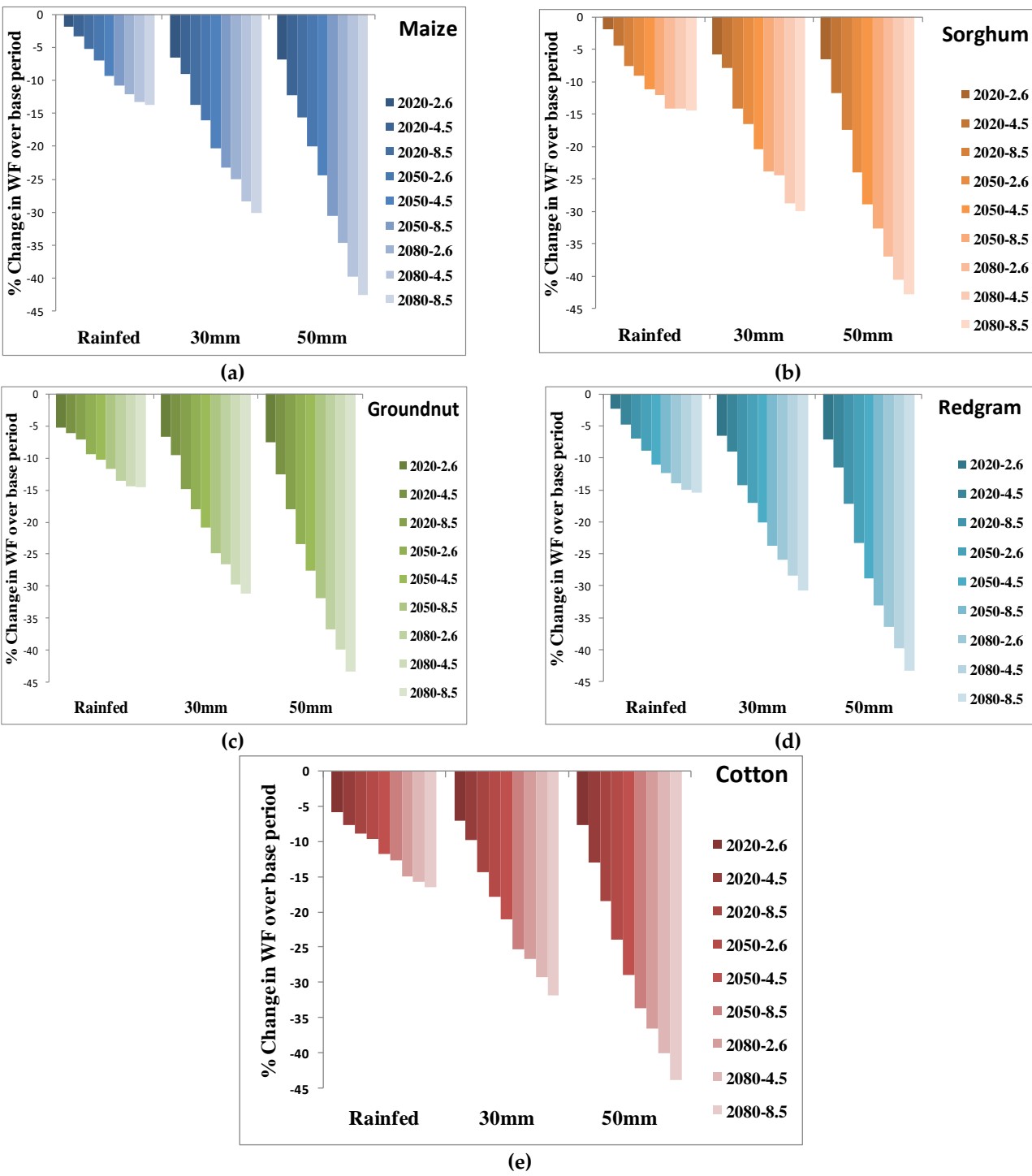

**Figure 6.** Percentage change in WF of rainfed crops ((**a**) maize (**b**) sorghum (**c**) groundnut (**d**) redgram and (**e**) cotton) with critical irrigation under different RCPs and time periods.

Groundnut

The groundnut yields taken for the WF analysis varied from 1.37 t/ha to 1.61 t/ha, 2.9 t/ha to 3.92 t/ha and 5.19 t/ha to 8.47 t/ha in rainfed, 30 mm and 50 mm CI, respectively,

in different RCPs and time periods (Table 3). The crop ET varied from 510 mm to 519 mm in RCP 2.6, 518 mm to 521 mm in RCP 4.5, and 513 mm to 533 mm in RCP 8.5 during the time period of 2020 to 2080. The ER varied from 320 mm to 329 mm, 322 mm to 331 mm, and 323 mm to 338 mm in RCP 2.6, 4.5, and 8.5, respectively, during the time period of 2020 to 2080 showing the increasing trend (Table 2). However, the ER was observed to be less than crop ET in all RCP and time periods.

The green WF under rainfed with ER contribution in the production varied from 2328 $m^3$/t to 2125 $m^3$/t, 2010 $m^3$/t to 2104.3 $m^3$/t and 2085 $m^3$/t to 2100$m^3$/t in RCP 2.6, 4.5 and 8.5, respectively for different time periods (Figure 5a).The blue WF with 30 mm CI varied from 207 $m^3$/t to 163 $m^3$/t, 201 $m^3$/t to 156 $m^3$/t and 189 $m^3$/t to 153 $m^3$/t in RCP 2.6, 4.5 and 8.5, respectively (Figure 5b). Similarly, with 50 mm CI, the blue WF varied from 193 $m^3$/t to 132 $m^3$/t, 182 $m^3$/t to 125 $m^3$/t, and 171$m^3$/t to 118 $m^3$/t (Figure 5c) in RCP 2.6, 4.5 and 8.5 during the time periods of 2020 to 2080. The percentage decrease in the green WF (rainfed) varied from 5.3–13.5%, 6.1–14.4% and 7.1–14.6% over the base period in different RCP and time periods (Figure 6c). At 30 mm CI, the blue WF varied from 6.7–26.6%, 9.6–29.7% and 14.8–31.1% over the base period in different RPCs and time periods. With a 50 mm CI strategy the blue WF varied from 7.5–36.7%, 12.6–39.9%, and 17.9–43.3% over the base period.

In time periods of 2020 to 2080 in different RCP's both green and blue WFs were decreased with maximum reduction in 50 mm CI strategy. In oilseed crops like groundnut, which is predominantly grown in south-central India is a profitable crop to the farmers with fewer WFs under the adaptation strategy.

Redgram

Redgram is a protein-rich leguminous crop which is commonly grown in rainfed conditions as a pulse crop. The yields of the crop varied from 0.82 t/ha to 1.0 t/ha, 1.6 t/ha to 2.18 t/ha, 2.8 t/ha to 4.58 t/ha in rainfed, 30 mm, and 50 mm CI strategies, respectively under different RCP's and time periods (Table 3). The crop ET varied from 457 mm to 462 mm, 458 mm to 463 mm, and 458 mm to 470 mm in RCP 2.6, 4.5, and 8.5 respectively in time periods of 2020 to 2080 with an increasing trend. The ER varied from 353 mm to 362 mm, 355 mm to 366, mm and 357 mm to 371 mm in different RCPs and time periods (Table 2). The ER was found to be less than $ET_c$ among all RCPs and time periods considered.

The green WF varied from 4308 $m^3$/t to 3795$m^3$/t, 4119 $m^3$/t to 3751$m^3$/t and 4104 $m^3$/t to 3728 $m^3$/t in RCP 2.6, 4.5 and 8.5, respectively during different time periods (Figure 5a). With a 30 mm CI strategy the blue WF varied from 371 $m^3$/t to 294 $m^3$/t, 361.5 $m^3$/t to 284 $m^3$/t and 341 $m^3$/t to 275 $m^3$/t in RCP 2.6, 4.5 and 8.5, respectively in the time period of 2020 to 2080 (Figure 5b). The blue WF with 50 mm CI varied from 357 $m^3$/t to 244.5 $m^3$/t, 340 $m^3$/t to 231.5 $m^3$/t and 318.5 $m^3$/t to 218 $m^3$/t in different RCP and time periods (Figure 5c). The percentage decrease in green WF varied from 2.4–14%, 4.8–15% and 7–15.5%, the percentage decrease in blue WF with 30mn CI varied from 6.6–26%, 9–28% and 14–30.7% by 2080, the percentage decrease in blue WF for 50 mm CI varied from 7.1–36.4%, 11.6–39.8% and 17.2–43.2% (Figure 6d) over the base period in different RCP and time periods.

Cotton

Cotton is grown by the farmers as a commercial crop in rainfed districts of south-central India having a growth period of 180 days. Its yields varied from 0.9 t/ha to 1.13 t/ha, 2.58 t/ha to 3.52 t/ha and 4.87 t/ha to 8.01 t/ha under rainfed, 30 mm CI and 50 mm CI, respectively for different RCP and time periods (Table 3). The crop ET varied from 707 mm to 716 mm, 709 mm to 718 mm, and 709 mm to 728 mm in RCP 2.6, 4.5, and 8.5 respectively in the time periods of 2020 to 2080. The ER varied from 402 mm to 409 mm, 402 mm to 414 mm and 403 mm to 424 mm under different RCPs and time periods. It was observed

that the ER was less than ET$_c$ among all RCPs and time periods (Table 2). Hence, ER was considered for calculating green WF under rainfed.

The green WF of cotton varied from 4202 m$^3$/t to 3795 m$^3$/t, 4120.4 m$^3$/t to 3765.5 m$^3$/t, and 4067.8 m$^3$/t to 3733 m$^3$/t in RCP 2.6, 4.5 and 8.5 for different time periods (Figure 5a). The blue WF with 30 mm CI varied from 233 m$^3$/t to 183.5 m$^3$/t, 225.6 m$^3$/t to 177 m$^3$/t and 214 m$^3$/t to 170.5 m$^3$/t in RCP 2.6, 4.5 and 8.5, respectively during 2020 to 2080 (Figure 5b). Adapting 50 mm CI in cotton reduced the blue WF over the 30 mm CI and rainfed. The blue WF with 50 mm CI varied from 205 m$^3$/t to 141 m$^3$/t, 193 m$^3$/t to 133 m$^3$/t and 181.5 m$^3$/t to 124.8 m$^3$/t (Figure 5c) in different RCP and time periods. The percentage decreases over the base period varied from 5.8% to 15%, 7.7% to 15.6% and 8.8% to 16.4%, with 30 mm CI it varied from 6.9% to 26.6%, 9.8% to 29.2% and 14.3% to 31.8% 1.6%, 2.4% and in 50 mm CI it varied from 7.6% to 36.5%, 13% to 40% and 18.3% to 43.8% (Figure 6e) in different RCP's and time periods. Cotton also has a decreasing trend in WF's over the time periods and climate change scenarios of RCP. However, with adaptation strategy of providing 30 mm and 50 mm CI reduced the WF due to increase in the yields with increased critical water use during the growth period of the crop.

## 4. Discussion

The commonly grown crops in SAT regions of India are Sorghum, maize, groundnut, redgram and cotton. Sorghum is grown extensively in both Indian and African SAT regions as it is a staple food for poor people. It has localized value additions as well as good fodder value for animals. Though maize is water-intensive crop, it is grown in rainfed regions extensively in most of the SAT regions due to its commercial value, used as feed and fodder to the animals and poultry. Groundnut and redgram are commercial oilseed and pulse crops, respectively that provides protein and in situ nitrogen fixation to the soil. Cotton is long duration commercial crop grown in 67% area in rainfed regions of India having a productivity of 200–275 kg/acre which is very low compared to the other cotton-growing countries. The other crops productivity ranges between 0.8–1.0 t/ha in SAT regions [3]. The above crops suffer from the water supplies during critical stages with long dry spells (30–45 days) due to rainfall breaks. It is seen from Figure 2 that the average temperature increase will be from 1–5 °C having a maximum in RCP8.5. Though rainfall has an increasing trend among RCPs and time periods from 2020–2080, there were more non-rainy days during crop growth period indicating more dryspells happened, calling for a scope of rainwater harvesting on farm for critical irrigation.

Water footprints were studied in SAT regions with critical irrigation strategies applying 30 mm and 50 mm two times during dryspells in the crop-growing period. The available water content for use by the crops in rainfed soils is about 100 mm/m [38]. If the depletion of available water content is not addressed during dryspells at critical stages of rainfed crops, the crop yields are reduced by 30–40% in different crops causing huge losses to the farmers. In order to minimize this loss besides enhancing the crop productivity, critical irrigation of 50 mm was provided by meeting the requirement of the crop during dryspell at 50% depletion and deficit irrigation with 30 mm. Taking these two points into consideration, water footprints of rainfed crops were estimated using both green and blue water. The positive climate effects on crop growth can be adjusted by effective rooting depth and nutrients by providing critical irrigations during dryspells which can improve water productivity by 20–40% [39].

Rainfed crops of maize and groundnut registered the lowest water footprint in all RCP scenarios. Blue water footprint of cotton/redgram was found to be highest in all RCP scenarios with either 30 mm or 50 mm CI while the lowest was recorded for groundnut. It was found that the response of the crop to the CI was positive realizing more yields in the rainfed regions [40].

Deep rootedness, as well as long duration of the crops (redgram and cotton) standing in the field, requires more green water for effective root spread and resource use in the root zone under rainfed conditions. However, the long crop duration/indeterminate nature

of the crop, makes it to survive and recoup from the extreme weather conditions. Crop growth cycle for these crops with a long duration of 180 days in the field suffers from moisture stress during the critical stage of pod/boll development and filling. The green WF was very high for such crops, if they are grown in rainfed conditions and it could be reduced with blue water supplies through CI reducing the WF of these crops. However, if the rain breaks occur immediately after seed germination, the crop suffers affecting the plant density and this stage becomes a critical stage for crop survival. Therefore, CI could be applied at any stage of the crop facing severe moisture stress, resulting in improvement of either crop stand or yields [41]. Under limited soil depths, shallow-rooted crops of groundnut and maize registered the lowest WF as the CI of 30 mm or 50 mm would make water available within root zone to improve crop yields. However, cotton crop, which is deep-rooted also could record lower WF at all RCP scenarios after groundnut which might be due to its deep rootedness and also due to high yields (nearly three times higher) over rainfed cotton crop without CI. Reduction in WF is possible with CI at all RCP scenarios by standardizing timing of irrigation, quantity, and method of irrigation which trigger the crop growth parameters and yield attributes to a greater extent in crops under rainfed SAT regions.

## 5. Conclusions

Water footprint assessment on watershed basis is required to select the most efficient cropping system per unit of water consumed, which ultimately results in not only conserving water but also economic benefits to the farmers through proper water resource development and use management, particularly in SAT regions. The present study deals with the assessment of water footprints of rainfed crops grown in the watershed with critical irrigation of 30 mm and 50 mm two times as an adaptation strategy to climate change. Out of the water balance of watershed obtained from SWAT, modeling indicated that there was an opportunity for water harvesting through On-Farm Reservoirs for critical irrigation in watersheds, as surface runoff increased due to an increase in the rainfall during the growing period of selected crops across RCP and increasing time periods. The crop yields were simulated using the AquaCrop model for both base period and climate change scenarios with two critical irrigations of 30 mm and 50 mm. The analysis of water footprints for rainfed crops on a watershed basis indicated that the lowest water footprint was observed in maize under the 50 mm CI strategy followed by groundnut, sorghum, redgram, and cotton. The strategy of 50 mm CI during two critical stages of the crops resulted in maximum reduction in the blue WF which is 6.6–37%, 12–40%, and 18–44% for RCP 2.6, 4.5, and 8.5, respectively among the selected crops. In the rainfed system with a green water footprint also resulted in the reduction of green water footprint across the RCP and the time period of 2020 to 2080 which is less than blue water footprint of the crops. It was the result of increasing rainfall in RCPs (1.2–24%) over the base period. Green WF could be reduced further by the application of organics or plastic mulches which needs further investigation and validation under field conditions. The present studies would help to bring a policy framework from governments to effectively use water and develop water-efficient crop plans for enhancing productivity in rainfed SAT regions.

**Supplementary Materials:** The following supporting information can be downloaded at: https://www.mdpi.com/article/10.3390/w14081206/s1, Figure S1: Input layers to SWAT model.

**Author Contributions:** K.S.R. (Konda Sreenivas Reddy): conceptualization, methodology and writing manuscript draft; V.M.: conceptualization, methodology; P.K.P.: review and editing; M.K., A.G.K.R. and P.: validation, formal analysis; M.P. and K.S.R. (Kotha Sammi Reddy): validation, data curation and supervision; V.K.S.: project administration, A.K.K.: formal analysis and investigation. All authors have read and agreed to the published version of the manuscript.

**Funding:** This research was funded by Agri-Consortium Research Platform on Water and National Innovations in Climate Resilient Agriculture (NICRA), Indian Council of Agricultural Research, New Delhi and The APC was funded by NICRA with grant no. 2-2(201)/17-18/NICRA.

**Institutional Review Board Statement:** Not applicable.

**Informed Consent Statement:** Not applicable.

**Data Availability Statement:** Data used in this study are duly available from the first authors on reasonable request.

**Acknowledgments:** The authors fully acknowledge the financial help received from ACRP-Water, ICAR and the climate data received from NICRA project for undertaking the present research. Also, we acknowledge the help, co-operation, and guidance received from the Directors of ICAR-CRIDA, Hyderabad, and ICAR-IIWM, Bhubaneswar in carrying out this research.

**Conflicts of Interest:** The authors declare no conflict of interest.

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
