# Peer review of "Water Footprint Assessment of Rainfed Crops with Critical Irrigation under Different Climate Change Scenarios in SAT Regions"

_water, doi:10.3390/w14081206_

Round 1

Reviewer 1 Report

This manuscript explores green and blue water footprints of rainfed crops for base period (1994 to 2013) and different climate change scenarios of 2020, 2050 and 2080, considering rainfed and two critical irrigations. The structure of the manuscript is clear and the method description is accurate. However, the research results lack further analysis, and there are many small mistakes in language expression.

Below are some detailed comments and suggestions:

  1. Abstract: the description of the research resultsin Abstract is too simple, it is suggested to enrich it.
  2. Line 47: “It indicated that the ER is less than crop ET under different RCPs and time periods and base period.” This sentence is not accurately expressed. Need to indicate that ER and crop ET is for crops’ growing period.
  3. In the abstract and text, SAT should be labeled with the full name Semi Arid Tropical (SAT) when it first appears. Then theabbreviation SAT is used in the following context.
  4. Line48: “RCP’s” should be corrected to “RCPs”. The same problem appears elsewhere in the paper. Please check it carefully.
  5. Lines 68-69: “SAT regions though contribute 60% of nutritive food grains is suffering with 20 to 35% undernourished population [5].”This sentence is inconsistent with the meaning of the above and the following. It’s better to delete it or put it in line 62.
  6. Line 84: “The major soil groups include Alfisols, Black soils and alluvial soils.”This sentence is abrupt in the paragraph. It’s better to be deleted.
  7. Lines 91-93: “At sub-national region level has calculated the WF of domestic, industrial and agricultural sectors, and at national level to the global level.”The grammar of this sentence is wrong. The sentence lacks a subject.
  8. Lines 93-99: “The green and blue......critically rainfall dependent.”This sentence is too long to read, so please break it up into two sentences.
  9. Line 102: remove the blank space in “foot prints”. Use the consistent expression "footprint" in the whole paper.
  10. Lines 109-110: this sentence lacks a subject. And the same problem occurs in lines 112-114.
  11. Line 233: the first letter of “Validation”need not to be capitalized.
  12. Figure 3: the meaning of PREC and SURQ should explained in the text or in the title of figure 3.
  13. Figure 4 lacks the grid linesof radar charts, so the research results cannot be clearly presented in the figure.
  14. Section 3.3 mainly explains the calculated data results and the variation characteristics of water footprint under different RCPs. The reasons for such changes in water footprint under different RCPs and different CI are not analyzed. Results and Discussion needs further analysisof the results.
  15. The conclusion should be enriched. What are the shortcomings of the research and what can be furtherstudied in the future?

Reviewer 2 Report

Manuscript ID: water-1643615
Type of manuscript: Concept Paper

Title: Water Footprint Assessment of Rainfed Crops with Critical Irrigation under different Climate Change Scenarios in SAT Regions

The subject of this study is falling within the general scope of the journal. The study was well planned and carried out. It is a study that will guide the use of water more effectively in agricultural areas and its publication will contribute to the practice. However, the following corrections should be made.

- The meanings of these terms should be explained in parentheses in the abstract

(SAT, SWAT, ER, RCP and WF)

 E.g; Water footprint (WF)

Reviewer 3 Report

1- Most abbreviations need to be briefed first in Abstract like SAT, RCP, ER, and ET. Also in line 39 add WF in brackets as a first abbreviation.

2- Add a future direction at the end of abstract

3- Line 56 requires related references

4- Lines 107-109, add appropriate references for SWAT and add about two sentences for prior studies of SWAT

5- Figure 1, the quality is too bad, please improve the quality and add coordinates (N, and E) to the map. 

6- Lines 153-147, where are these maps? Please add them in the supplementary file

7- Add another Figure for the climatic data under the studied scenarios. In this case, the authors should add another subtitle in Methods called climatic data.

8-  Discussion is too weak, please enhance it with suitable recent references

Round 2

Reviewer 3 Report

No further comments.